# Structural Basis for the Inhibition of the Autophosphorylation Activity of HK853 by Luteolin

**DOI:** 10.3390/molecules24050933

**Published:** 2019-03-07

**Authors:** Yuan Zhou, Liqun Huang, Shixia Ji, Shi Hou, Liang Luo, Conggang Li, Maili Liu, Yixiang Liu, Ling Jiang

**Affiliations:** 1Key Laboratory of Magnetic Resonance in Biological Systems, State Key Laboratory of Magnetic Resonance and Atomic and Molecular Physics, National Center of Magnetic Resonance in Wuhan, Wuhan Institute of Physics and Mathematics, Chinese Academy of Sciences, Wuhan 430071, China; 15044164285@163.com (Y.Z.); huangliqun15@mails.ucas.ac.cn (L.H.); jishixia@gmail.com (S.J.); luoliang17@mails.ucas.ac.cn (L.L.); conggangli@wipm.ac.cn (C.L.); ml.liu@wipm.ac.cn (M.L.); 2Graduate University of Chinese Academy of Science, Beijing 100049, China; 3Laboratory of Computer-Aided Drug Design and Discovery, Beijing Institute of Pharmacology and Toxicology, Beijing 100850, China; houshi28@126.com

**Keywords:** two-component system, histidine kinase, luteolin, ATP, NMR, molecular docking, inhibition

## Abstract

The two-component system (TCS) is a significant signal transduction system for bacteria to adapt to complicated and variable environments, and thus has recently been regarded as a novel target for developing antibacterial agents. The natural product luteolin (Lut) can inhibit the autophosphorylation activity of the typical histidine kinase (HK) HK853 from *Thermotoga maritime*, but the inhibition mechanism is not known. Herein, we report on the binding mechanism of a typical flavone with HK853 by using solution NMR spectroscopy, isothermal titration calorimetry (ITC), and molecular docking. We show that luteolin inhibits the activity of HK853 by occupying the binding pocket of adenosine diphosphate (ADP) through hydrogen bonds and π-π stacking interaction structurally. Our results reveal a detailed mechanism for the inhibition of flavones and observe the conformational and dynamics changes of HK. These results should provide a feasible approach for antibacterial agent design from the view of the histidine kinases.

## 1. Introduction

Bacterial resistance caused by antibiotics misuse has become a public health problem affecting human health [1,2]. Therefore, the research and development of new antibacterial agents have drawn more and more attention [3,4]. In recent years, many researchers have focused on a potential drug target, two-component systems (TCSs) [5]. TCSs are the major pathways by which bacteria sense environmental stimuli and initiate cellular responses [6,7]. TCS proteins participate in regulating cell division, osmotic equilibrium, spore formation, and other life activities of bacteria [8]. Recent studies have also found that TCSs are directly involved in bacterial virulence [9,10]. TCSs have similar structural characteristics and a relatively conservative signal transduction mechanism [11]. TCSs exist widely in bacteria but have never been found in mammalian cells [12]. Therefore, the development of novel antibacterial agents targeting TCSs has become a promising research direction [13].

A typical TCS is composed of two proteins, histidine kinase (HK) and the response regulator (RR). TCSs transmit signals from the extracellular environment to bacteria via phosphotransfer [14]. As shown in Figure 1a, the extracellular sensing domain of HK senses environmental stimuli to propagate conformation changes, and then the cytoplasmic portion is autophosphorylated. P~HK (phosphorylated HK) transfers its phosphoryl group to the downstream cognate RR. After being phosphorylated by HK, P~RR (phosphorylated RR) binds to genes or other proteins to respond to the environment stimuli [15]. Adenosine triphosphate (ATP) provides the original phosphate group for HK and participates in this process together with Mg^2+^ [16]. HK853 is a typical HK expressed by gene *TM0853* of *Thermotoga maritime*, whose cognate response regulator is RR468 [17,18,19,20]. The cytoplasmic portion of HK853 (HK853^cp^) plays a predominant role in the transphosphorylation. Structurally, HK853^cp^ is composed of two domains, N-terminal histidine-containing phosphotransfer domain (HK853^DHp^) and C-terminal catalytic and ATP-binding domain (HK853^CA^) [15] (see Figure 1b). HK853^CA^ is the binding site of ATP/adenosine diphosphate (ADP) and is an important starting point for signal transduction.

Flavones are a class of natural botanicals widely distributed in plants, and most flavones have a typical 2-phenychromen-4-one structure as a parent nucleus. Various functional groups such as phenolic hydroxyl, methyl, and methoxyl groups link to different sides of the parent nucleus to form different kinds of flavones. Traditional Chinese medicine contains plentiful flavones as active ingredients, such as luteolin (Lut), quercetin (Que), and apigenin (Api) [21,22,23]. The structure of Lut contains four hydroxyl groups at 5, 7, 3′, and 4′ sites (see Figure 1c). Current research indicates that Lut has anti-inflammation, antioxidant, antineoplastic, and antibiosis effects [24,25,26]. It has also been demonstrated that Lut can significantly inhibit the activity of methicillin-resistant staphylococcus aureus [27,28]. About 53,000 diverse small molecules have been screened for the ATP-binding pocket of HK by fluorescence polarization displacement assay [13]. As one of the active substances based on screening results, Lut has shown significant antimicrobial activity according to antimicrobial tests. It has also been reported that Lut and other flavones can inhibit kinase enzymes [29,30,31,32]. For example, Lut can bind and inhibit a ser/thr protein kinase, casein kinase 2 (CK2), which is related to gene expression, protein synthesis degradation, and whole cell cycle [33]. Another ser/thr protein kinase, death-associated protein kinase 1 (DAPK1), is also a receptor for Lut [34]. However, the specific mechanism and the role of the binding effect on histidine kinase from TCSs by Lut are still unclear.

Using a combined NMR and isothermal titration calorimetry (ITC) techniques, along with molecular docking, we herein investigate the binding mechanism between HK853 and Lut, and analyze the structural details of HK853 bound with Lut. The inhibition effects of another four flavones have also been assayed. Our results reveal the binding mechanism of flavones with histidine kinases, and also provide relevant information on the antibacterial drug design targeting TCSs.

## 2. Results and Discussions

### 2.1. Inhibition of the Autophosphorylation Activity of HKs by Lut

The autophosphorylation of HK is the first step of signal transduction mediated by phosphoryl group transferation. In this step, ATP binds to the CA domain of HK and then is converted to adenosine diphosphate (ADP) and a phosphoryl group. However, the phosphoryl group which is bound to the conserved histidine residue of HK has a short lifetime and is rapidly released to form a free phosphate in solution.

Since NMR has particular advantages in tracking dynamic reactions in solution, we used NMR spectroscopy to observe the entire reaction process. The ^31^P nucleus has a high natural abundance and the ^31^P-NMR spectra of ATP and ADP are well resolved. Hence, we carried out ^31^P-NMR experiments to assign all the phosphorus atoms of the products in the reactions (see Appendix A), and then quantitatively monitored the changes of ATP and its degradation products to characterize the catalytic activity of HK.

HK853 from *Thermotoga maritime*, which was extensively studied with plenty of structural information, was used as a model histidine kinase protein [35,36,37]. Accordingly, a higher temperature of 40 °C was used as the reaction conditions, and two groups of samples were prepared to monitor the reaction between the cytoplasmic portion of HK853 (HK853^cp^) and excessive ATP in the presence and absence of Lut, respectively. In the absence of kinase, ATP/ADP barely degrades at 40 °C (see Appendix A). When HK853^cp^ was introduced, the peak intensities of ATP decrease rapidly as the reaction time increases, while those of ADP and Pi are enhanced. After the peak integration, as shown in Figure 2, we observed that in the presence of Lut, the residual proportions of ATP are greater than the data in the absence of Lut after 4 h. After 8 h, ATP in the samples without Lut is completely hydrolyzed by HK853^cp^, while the Lut containing group still shows about 10% remaining ATP. The results indicated that Lut inhibits the autophosphorylation activity of HK853^cp^. At the same time, another histidine kinase, EnvZ, was chosen to verify the inhibition. Similar to HK853^cp^, the autophosphorylation of EnvZ is also significantly inhibited by Lut, with an even stronger inhibition activity under the reaction conditions at 25 °C (see Appendix A). Therefore, our results verified the inhibitory effect of Lut on the autophosphorylation activity of HKs by direct observation of the changes of ATP and ADP.

### 2.2. Determination of the Binding Affinities of Lut/ADP to HK853^cp^

To explore the inhibition mechanism of Lut, isothermal titration calorimetry (ITC) experiments were used to detect the dissociation constant (K_d_) between Lut and the kinase. According to the published crystal structure (see Figure 1b), ADP binds to the cytoplasmic portion of HK853 (HK853^cp^) [15]. HK853^cp^ was titrated with Lut and ADP respectively, and the results of titration experiments are shown in Figure 3. Both ADP and Lut are bound to HK853^cp^ at a ratio of 1:1, but the binding affinities of ADP and Lut are relatively weak. Unexpectedly, the binding affinity of Lut is about 10 times weaker than that of ADP. The K_d_ of Lut/HK853^cp^ is 187.3 ± 7.7 μmol, while that of ADP/HK853^cp^ is 19.0 ± 1.3 μmol. Since the signal transduction process is transient in most cases, the binding and dissociation of ATP/ADP with HK are required to ensure successful delivery of phosphate groups. Thus, the weak binding affinity of ADP to HK is reasonable. Interestingly, although the ITC experiment indicates that the binding affinity of Lut is weak, Lut still inhibits the activity of HK853, so it is necessary to explore the detailed inhibition mechanism at the atomic level.

### 2.3. Detection of the Binding Site and Conformational Changes

The CA domain of HK853 (HK853^CA^) is the binding site of ATP/ADP according to the published complex structures [15]. We constructed and labeled the CA domain and assigned the signals of its ^15^N-^1^H HSQC spectrum through the triple resonance experiments. Structures of proteins are crucial for their biological functions [38,39,40]. Functional changes are often accompanied with a conformational change or switch [41,42,43,44]. NMR experiments have their unique advantages in detecting the conformation and dynamics changes and searching for the binding sites at the atomic level. In order to obtain the binding information at the atomic level, we conducted NMR titration experiments of HK853^CA^ with Lut (Figure 4a), and carefully compared the chemical shift perturbations of the peaks before and after Lut binding, and also the changes in signal intensities, and then summarized them in Figure 5. Basically, we found that these residues have undergone strong changes mainly in three regions. As shown in Figure 5, the mostly perturbed residues are located in the ADP binding pocket mapped in blue on the crystal structure [15]. Therefore, we speculated that Lut occupies the same spatial position as ADP.

To further investigate the conformational change after Lut binding, we performed an ADP titration experiment. When ADP is bound with HK853^CA^, the complex presents a new set of signals, showing the slow exchange property at the NMR time scale, and the chemical shift changes disperse in the spectrum indicating a large perturbation of the protein structure (Figure 4b). Unlike the binding of ADP, about one-third of the peaks of HK853^CA^ have almost no changes after Lut binding (Figure 4a). Additionally, some signals of the Lut-bound complex are weakened, as compared in Figure 4c–e. These three peaks are from different locations of HK853^CA^. After binding to Lut, the peak intensities of Thr436 and Gly469 are significantly attenuated. However, the intensity of Gly381 barely changed. These differences indicate that ADP binding is more stable than Lut, which is consistent with the ITC results.

By carefully analyzing these weakened and disappeared residues (see Figure 5), we found that many of the peaks are located in the β5-α3 loop, α3-α4 loop, and β6-β7 loop of the CA domain, indicating that the binding of Lut changed the dynamics of these loops. Among these loops, the key ATP lid in the α3-α4 loop is known to play a crucial role for kinase autophosphorlyation [15,45]. We also analyzed the significant changes of the other residues, and found that they are mainly distributed in four functional areas named N-box, G1-box, F-box, and G2-box that are reported in the literature [5]. More notably, according to previous studies, Arg430 was an important residue for the autophosphorylation of HK853^cp^ [46]. Our results indicate that Lut has a significant effect on Arg430, since the Arg430 peak disappears after Lut titration (see Figure 5). These results further confirm that the binding position of Lut is located in the same binding pocket of ADP.

In summary, our results indicate that Lut competes with the ADP binding pocket of HK853^CA^ to inhibit the autophosphorylation activity of HK853, and its binding causes conformational and dynamics changes of several major functional regions of the CA domain.

### 2.4. Structural Insights of Lut Inhibition Suggested by Molecular Docking

To obtain structural insights, molecular docking was performed by using Accelrys Discovery Studio Client 2.5. Lut was regarded as the ligand and HK853^cp^ was the receptor protein. By searching for the most stable complex structure, a detailed binding message, such as binding sites and acting forces would be acquired. The docking result is shown in Figure 6, where Lut is wrapped in a cavity constituted by Tyr384, Asp411, Gly413, Ile414, Gly415, Ile416, Tyr429, Arg430, Val431, Asp432, Ser433, Thr436, Gly469, Ser470, and Phe472. Ten hydrogen bonds (H-bonds) stabilized the complex structure together. The 4′-hydroxyl group of Lut forms four H-bonds respectively with side chain oxygen atoms from Asp411 and Ser470, and the amide protons from Ile414 and Gly415. Additionally, the above two oxygen atoms also form two H-bonds with the 3′-hydroxyl hydrogen atom of Lut. The 5-hydroxyl oxygen atom of Lut and the amide proton of Val431 form another H-bond. The other three H-bonds are formed by the 7-hydroxyl of Lut and the oxygen atom of backbone carbonyl from Val431, and oxygen atoms of the backbone and side chain from Thr436, respectively. A π-π stacking interaction is also observed between Lut and Tyr384, which is maintained by two aromatic rings.

This result is highly consistent with the NMR data. In the titration NMR spectrum, the chemical shifts of Tyr384, Ile414, Ile416, Tyr429, and Thr436 change significantly. After Lut binding, peaks of Asp411, Gly413, Gly415, Arg430, Gly469, and Phe472 disappear (see Figure 5). The two sets of results confirm each other. By comparing the docking structure in Figure 6b with the crystal structure in Figure 6c, Lut and ADP occupy the same spatial position, which is consistent with our speculation in Section 2.3. In addition, ^15^N-^1^H HSQC competition experiment results (see Appendix A) further support this conclusion.

### 2.5. Verification of the Inhibition Mechanism by Other Flavones

It has been revealed that ADP interacts with HK mainly through hydrogen bonds [15]. Similar with ADP, Lut has its oxygen atoms on rings and hydroxyl side chains. Thus, it can be inferred that other flavones which have similar chemical constructions as Lut might have similar inhibition effects. We carried out the same experiments to study the inhibition effects of other four flavones on the autophosphorylation activity of HK853^cp^. These four flavones all have at least two critical hydroxyl groups, and their chemical constructions are shown in Figure 7a. In these experiments, all the conditions are the same as those of Lut investigations. The ATP ratio recorded at different reaction times was shown in Figure 7b. According to the results shown in Figure 7b, the reaction rates of the HK autophosphorylation are reduced by the four flavones, indicating that all these four flavones can inhibit the activity of HK853^cp^.

To verify the similar binding mechanism of these four flavones, Api and Kae were chosen for ^15^N-^1^H HSQC titration experiments. These four flavones were chosen because it is suggested that their hydroxyl groups are significant for ligand binding. As shown in Appendix A, Api and Kae both bind to HK853^CA^, and the spectra of the complexes are similar as the spectrum of the Lut titration sample.

Taken together, the highly conserved ATP-binding site of HK is considered to be a promising target for broad-spectrum antibacterial drugs. The inhibition activities of Lut and other flavones were investigated by NMR, ITC, and molecular docking experiments. Limitations still exist for these ligands, such as weak binding affinity and high inhibition concentration. Some of them might not be suitable for the future drug design. However, our research provides information on the conformational and dynamics changes of HK after ligand binding, and suggests the essential residues of functional loops involved in the ligand binding. These results should shed light on the design of inhibitors from the view of the histidine kinases.

## 3. Materials and Methods

### 3.1. Reagents

Luteolin (HPLC ≥ 98%), apigenin (HPLC ≥ 98%), kaempferide (HPLC ≥ 98%), 5,7,3′,4′,5′-pentahydroxyflavone (flavone 3, HPLC ≥ 98%), and 7,8,3′,4′-tetradroxyflavone (flavone 4, HPLC ≥ 98%) were all purchased from PureOne Biotechnology (Shanghai, China). ATP (HPLC ≥ 98%), ADP (HPLC ≥ 98%), and AMP (HPLC ≥ 98%) were purchased from Sigma (Shanghai, China).

### 3.2. Protein Preparation and Purification

Plasmids encoding HK853^cp^ (residues 232–489) and HK853^CA^ (residues 319–489) were transformed into *E. coli* BL21(DE3) gold cells (Novagen). For unlabeled HK853^cp^, the transformed bacterial cells were cultivated in the LB medium containing ampicillin (100 μg/mL) at 37 °C. When OD_600_ of the culture reached to 0.7~0.8, the cells were induced with IPTG (isopropyl β-d-1-thiogalactopyranoside, 1 mM) at 20 °C for 20 h. ^15^N-labeling and ^13^C-labeling of HK853^CA^ was achieved by growing cells in the M9 minimal medium containing ^15^N-labeled NH_4_Cl and ^13^C-labeled d-glucose (Cambridge Isotope Laboratories, Andover, MA, USA). Other cultured and induced conditions of cells were the same as those of HK853^cp^.

HK853^cp^ and HK853^CA^ were both purified by ammonium sulfate precipitation (50% *w*/*v*), followed by anion-exchange and size-exclusion chromatography. The anion-exchange buffer contained 20 mM Tris and the size-exclusion chromatography buffer contained 20 mM Tris, 200 mM NaCl. Both buffers were adjusted to pH 8.5.

### 3.3. ITC Experiments

Calorimetric titrations were performed on an iTC200 microcalorimeter (GE). Protein samples were extensively dialyzed against the ITC buffer containing 20 mM Tris (pH 8.0) at 25 °C. The sample cell was typically filled with 15 µM of protein, and the injection syringe was filled with 200 µM of Lut. Lut was initially dissolved into the alkalinized ITC buffer and then the Lut solution was readjusted to pH 8.0, because the Lut has higher solubility in alkaline solutions. The titration typically conducted by a preliminary injection and 24 subsequent injections. Data for the preliminary injection were discarded. The data were analyzed with Origin 7.0.

### 3.4. NMR Experiments

• ^31^P-NMR experiments

NMR data were collected on a Bruker 600-MHZ spectrometer and analyzed by MestReNova. The spectra were acquired with 1340-2680 scans. HK853^cp^ sample, Lut, and ATP powder (Sigma) were dissolved in the NMR buffer containing 20 mM Tris, 50 mM KCl, and 10 mM MgCl_2_, with 10% D_2_O (pH 8.0). HK853^cp^, Lut, and ATP solutions were mixed to make up the reaction solution and the final concentration ratio was 1:2:20. The concentration of HK853^cp^ was 0.4 mM. The ratio of ATP was calculated by this equation: ⅓S_ATP_/(⅓S_ATP_ + ½S_ADP_ + S_AMP_ + S_Pi_) × 100% (S means integral area of peaks, e.g., S_ATP_ means the integral area of three ATP peaks).

• ^15^N-^1^H HSQC NMR experiments

NMR data were collected on a Bruker 700-MHZ spectrometer and analyzed by TopSpin 3.2. The 2D ^15^N-^1^H HSQC spectra were acquired with 24 scans with 1024 and 128 data points for ^1^H- and ^15^N-dimensions, respectively. The chemical shift perturbations for backbone ^1^H and ^15^N resonances were calculated by this equation: (∆^1^H^2^ + (0.2 × ∆^15^N)^2^)^0.5^. ^15^N-HK853^CA^ sample, Lut, and ATP powder (Sigma) were dissolved in the NMR buffer containing 20 mM HEPES, 50 mM KCl, and 10 mM MgCl_2_, with 10% D_2_O (pH 7.0).

## 4. Conclusions

HKs are considered to be a novel drug target for antibacterial agents. We verified the inhibition effect of Lut to the autophosphorylation activity of HKs, and revealed the binding mechanism of this effect through combined NMR, ITC, and molecular docking studies. Structurally, Lut could bind to HK853^CA^ by hydrogen bond forces and π-π stacking interaction. In the complex structure, Lut occupies the binding pocket of ADP, and causes conformational changes of the crucial functional regions of HK: ATP-lid, N-box, G1-box, F-box, and G2-box, including an impacted catalytic residue Arg430. The inhibition mechanism has been further verified by another four flavones.

We suggest that the subsequent drug design and screening might take the dynamics of the target protein into account, and also the interactions between ligands and relevant residues. Since the active site of histidine kinases is conserved but different from other enzymes, our results provide not only the structural binding site but also the dynamics changes of some critical loops, which will inform the design of better inhibitors specifically targeting histidine kinases.

## Figures and Tables

**Figure 1 molecules-24-00933-f001:**
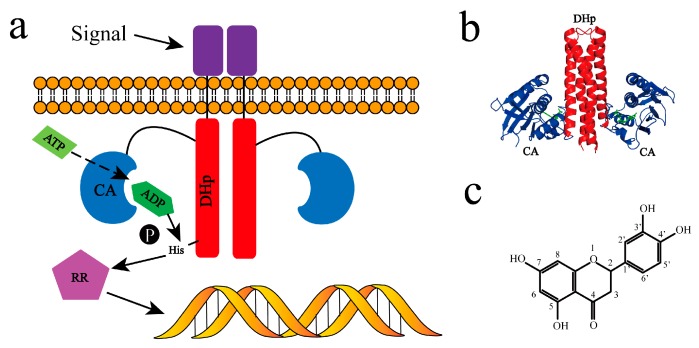
Typical two-component system HK–RR signaling pathway and the potential inhibitor luteolin (Lut). (**a**) The detailed process of signal transduction via histidine kinase (HK) and the downstream response regulator (RR); (**b**) crystal structure of HK853^cp^-ADP complex [15]. HK853^DHp^ was colored in red, HK853^CA^ in blue, and the two adenosine diphosphate (ADP) molecules in green; (**c**) chemical construction of luteolin.

**Figure 2 molecules-24-00933-f002:**
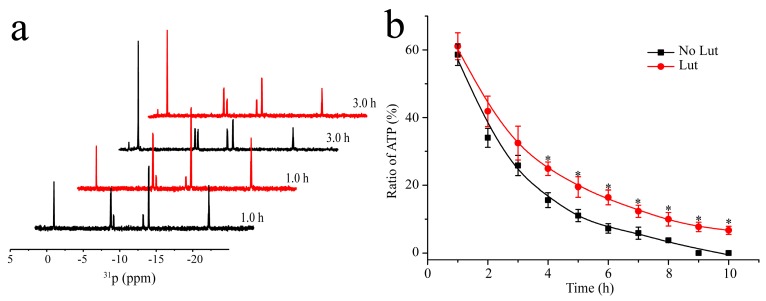
Inhibition of the autophosphorylation activity of HK853^cp^ by luteolin. (**a**) ^31^P-NMR spectra of the reaction between HK853^cp^ and ATP (reaction conditions: 20 mM Tris, 50 mM KCl, 10 mM MgCl_2_, 8 mM ATP, 0.4 mM HK853^cp^, and 0.8 mM Lut (in Lut group) with 10% D_2_O, pH 8.0) reaction time of 1 h and 3 h. The reaction groups in the absence of luteolin are shown in black, while the reaction groups in the presence of luteolin are in red; (**b**) the ratio of ATP in the reaction mixture obtained by the integration of ^31^P-NMR signals. The no-luteolin group was labeled by ■. The luteolin containing group was labeled by ●. (*n* = 3, * *p* < 0.01, compared to no-luteolin group).

**Figure 3 molecules-24-00933-f003:**
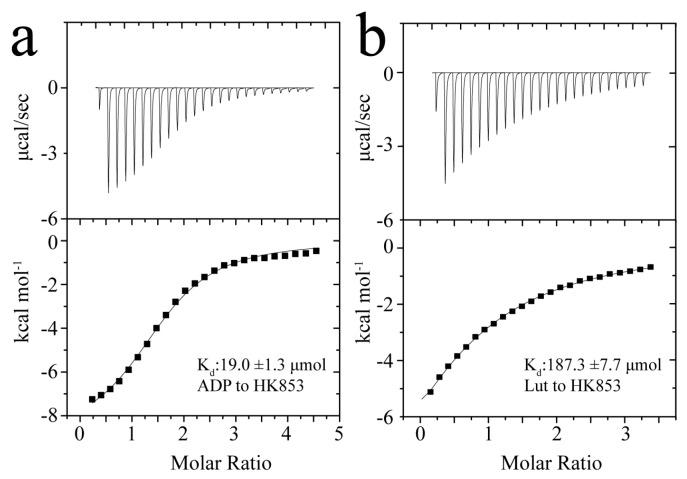
Binding affinities between HK853^cp^ and ADP/Lut. (**a**) The result of ADP–HK853^cp^ binding reaction; (**b**) the results of Lut–HK853^cp^ binding reaction.

**Figure 4 molecules-24-00933-f004:**
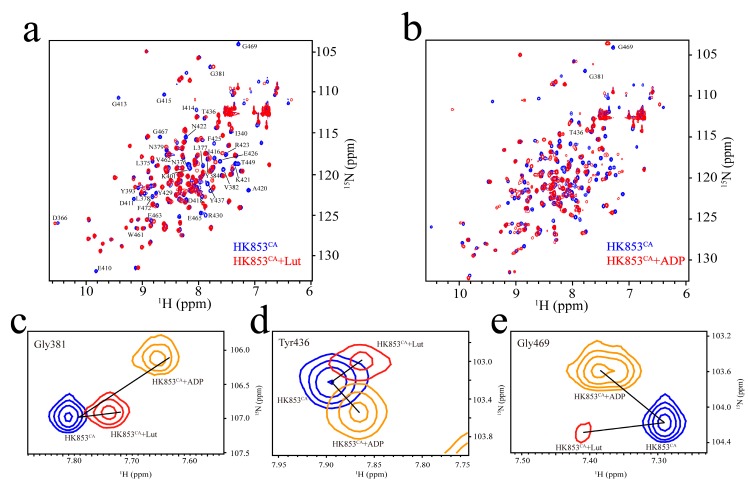
Interactions between HK853^CA^ and ADP/Lut. (**a**) The spectrum in blue shows the ^15^N-^1^H HSQC signals of HK853^CA^ without Lut, while the Lut containing group is in red (reaction conditions: 20 mM HEPES, 50 mM KCl, 10 mM MgCl_2_, 0.4 mM HK853^CA^, and 0.8 mM Lut with 10% D_2_O, pH 7.0); (**b**) the spectrum in blue shows the ^15^N-^1^H HSQC signals of HK853^CA^ without ADP, while the ADP containing group is in red (reaction conditions: 20 mM HEPES, 50 mM KCl, 10 mM MgCl_2_, 0.4 mM HK853^CA^, and 0.8 mM ATP with 10% D_2_O, pH 7.0); (**c**) the superimposed spectra of peak Gly381. Peaks from spectrum HK853^CA^, HK853^CA^ + ADP and HK853^CA^ + Lut were respectively exhibited in blue, orange, and red; (**d**) the superimposed spectra of peak Tyr436, with the same color code as (**c**); (**e**) the superimposed spectra of peak Gly469, with the same color code as (**c**).

**Figure 5 molecules-24-00933-f005:**
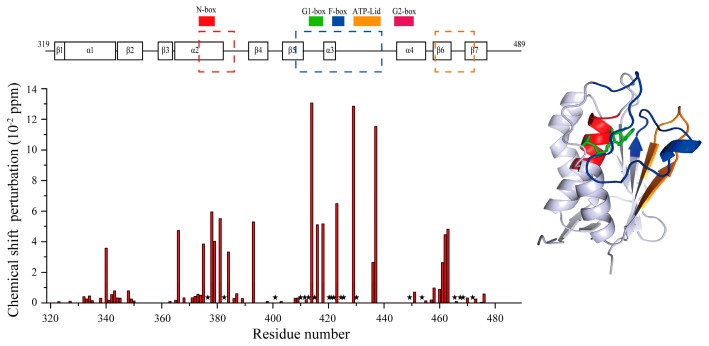
Chemical shift perturbation of residues after Lut binding. Disappeared signals are represented by ★. Obviously changed regions are respectively colored by red, blue and orange.

**Figure 6 molecules-24-00933-f006:**
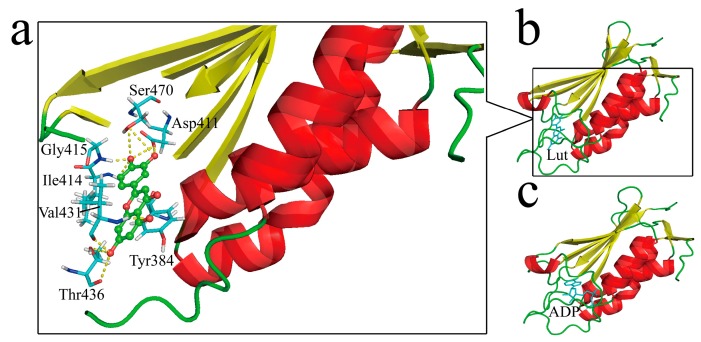
Structural insights of Lut binding to HK853^CA^: (**a**) Interaction details of Lut with the surrounding residues of HK853^CA^; (**b**) simulative secondary structure of Lut binding to HK853^CA^; (**c**) crystal structure of ADP bound HK853^CA^ [15].

**Figure 7 molecules-24-00933-f007:**
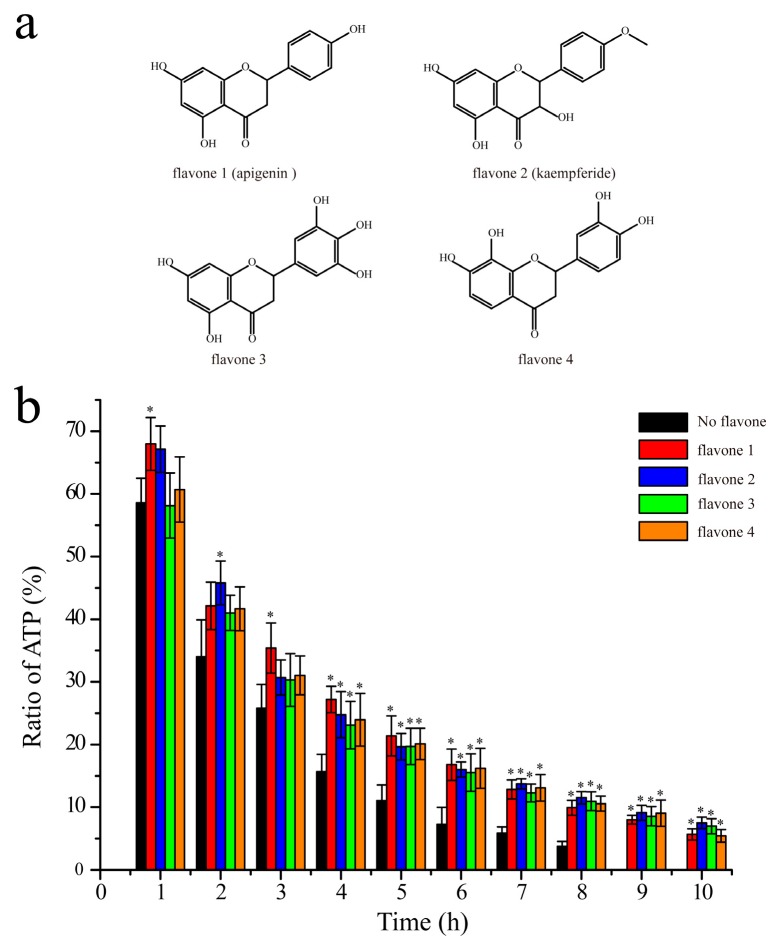
Inhibition of the autophosphorylation activity of HK853^cp^ by other flavones: (**a**) Chemical constructions of flavones 1, 2, 3, and 4; (**b**) the inhibition data are shown in red, blue, green, and orange for flavones 1, 2, 3, and 4, respectively (reaction conditions: 20 mM Tris, 50 mM KCl, 10 mM MgCl_2_, 8 mM ATP, 0.4 mM HK853^cp^, and 0.8 mM flavone with 10% D_2_O, pH 7.0). The control group without flavones is colored in black (*n* = 3, * *p* < 0.01 compared to the group without flavone).

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
