# Peer review of "Structural Basis for the Inhibition of the Autophosphorylation Activity of HK853 by Luteolin"

_molecules, 2019, doi:10.3390/molecules24050933_

Round 1

Reviewer 1 Report

The authors have done a careful study to provide structural basis for the inhibition of the autophosphorylation activity of HK853 by Luteolin. Overall I find this manuscript to be of low significance because the inhibition of HK853 as well as bacterial growth by luteolin is weak so one wonders the justification of spending such resources to study the binding, unless the study is going to inform the design the synthesis of more potent analogs. In that regard, the manuscript would have been considered important if the authors had used the information to make better compounds.

In addition to lack of significance, here are some other comments that the authors should address:

1) Figure legends should indicate how much compound is being used. For example in Figures 2, 4, 7.

2) In Table 1, the concentration of compounds should be indicated. From the experimental section, it appears that 0.3 mM (or 300 uM) of compound was used and even at such high concentration, the inhibition is modest. 

3) Critical controls:

"By comparing the docking structure in Figure 6b with the crystal structure in Figure 6c, Lut and ADP occupy the same spatial position which is consistent with our speculation in section 2.3."

There are more sophisticated and better methods to do that will provide better evidence that the two molecules occupy the same spatial position than what is provided in the manuscript. For example fluorescent-labeled ADP can be used to get apparent Kd (using Fluorescence polarization). One can also use the ITC for Kd for labeled ADP. Then labeled ADP can be competed against with Lutein to get binding parameters and analyzed to determine if the two are competing for the same site. There are many models that will allow the authors to do this.

Such experiment can add more insights to the NMR experiments to provide a more complete picture.

A better justification to the work has to be provided. Why is this important to do in light of the very weak potency of compound tested? How does this study inform the design of better compounds? What are some of the limitations of the tested compounds (pharmacologically, considering that it has many phenolic moieties, which are not desired in a drug). Can better analogs be made, based on the initial data (design of research methods)? These have to be discussed in an honest fashion to make the work publishable.

Author Response

We appreciate for your remindings and suggestings. A potint-by-point response to your comments has been uploaded as a Word file.

Reviewer 2 Report

This paper demonstrated that the natural product luteolin can inhibit the activity of HK853 by occupying the binding pocket of ADP via hydrogen bonding and π-π stacking interactions and suggested that the optimization of flavones and their analogues can be a promising approach for antibacterial therapeutics. The methods and results are sound. However, several major points should be addressed.

Major points:

To identify that Lut can inhibit the autophosphorylation activity of HK853, the authors performed NMR spectroscopy to observe the changes of ATP and its degradation products to characterize the catalytic activity of HK. However, ATP takes parts in many other enzyme reactions, this assay does not show that the natural product can inhibit HK853 selectivity. It is recommended to perform other methods to further verify the results. 

In the following experiments, the authors performed ITC to determine the binding ability of the natural products to HK853. The results indicated that there is a weak binding affinity of Lut. I wonder whether there are other proteins that can target Lut. It is recommended to perform ITC to measure other HK related proteins for their binding affinity to Lut.

In section 2.5, the authors detected the inhibition ability of other flavones on HK853. The results indicated that all four flavonoids can inhibit the activity of HK853. Since many flavonoids show low selectivity, selectivity experiments are required to evaluate these flavones for HK activity inhibition.

In table 1, the authors showed the survival rate of these four flavones, however, the data revealed that all of these had less inhibition on BL21 gold cells. In addition, no concentration and dosage information were shown in the table, please add it.

The authors used BL21 gold cells as the model cell line to evaluate the inhibition ability of these compounds. The reason for choosing BL21 Gold cells need to be described, instead of simply showing the reference.

In the introduction, the authors have stated that flavonoids including Lut have anti-inflammation, antioxidant, antineoplastic and antibiosis effects, and particularly the effect of Lut on antimicrobial activity. However, no references showing the potential inhibition of Lut on kinases were provided. It is recommended to introduce the activity of flavonoids on kinase enzymes and describe the reason of choosing HKs as the potential targets for Lut. The following references for potential kinase inhibition by flavonoids may be cited: 

1. Chemistry–An Asian Journal, 2018, 13(3): 275-279.2. 

2. Anti-Cancer Agents in Medicinal Chemistry (Formerly Current Medicinal Chemistry-Anti-Cancer Agents), 2013, 13(3): 456-463.

3. Methods, 2015, 71: 38-43.

4. Proceedings of the National Academy of Sciences, 2000, 97(24): 13330-13335.

Overall, I recommend publication of this manuscript after major revision.

Author Response

We appreciate for your remiandings and suggestions. And we have uploaded a point-by-point response to your comments as a Word file.

Round 2

Reviewer 1 Report

Revised manuscript is better and clarifies that this is just mechanistic work, based on the fact that the inhibition/binding is weak. Authors now discuss limitations.

Reviewer 2 Report

The authors have satisfactorily responded to all the questions and made the necessary changes to the manuscript. I have no further questions and suggest the acceptance of the revised manuscript.